# An Experimental and Numerical Study on Impact and Compression after Impact of Stiffened Composite Panels

**DOI:** 10.3390/polym15010165

**Published:** 2022-12-29

**Authors:** Peiyan Wang, Yongshun Chen, Chunxia Yue, Wei Zhao, Chenchen Lian, Ke Zhang, Jie Zheng, Zhufeng Yue

**Affiliations:** 1Department of Engineering Mechanics, Northwestern Polytechnical University, Xi’an 710129, China; 2AVIC Xi’an Aircraft Industry Group Company Ltd., Xi’an 710089, China; 3AVIC the First Aircraft Institute, Xi’an 710089, China

**Keywords:** stiffened composite panel, low-velocity impact, fatigue load, CAI, Hashin criterion

## Abstract

To develop the full application potential of composite materials, research on the post-buckling behavior of composite stiffened panels is of great significance. In this paper, the impact and compression after impact (CAI) behaviors of four different types of composite stiffened panels were studied by numerical simulation and experimental methods. The low-velocity impact damage simulated dynamically was introduced as the initial state in the compression simulation, and a two-dimensional shell model with Hashin failure criteria and stiffness degradation was adopted to estimate the failure load of composite stiffened panels under impact and CAI. The error between simulation results and test results was less than 10%, showing that the method used in this study achieved considerable accuracy in experimental results. Analysis of the impact test results revealed that the extent of damage is related to many factors, including the cross-sectional size of stiffeners, the spacing of stiffeners, and the material and thickness of the skin. In addition, the influence of fatigue damage on residual strength after impact was also studied experimentally, with results showing that the buckling and failure loads decreased by about 5% under 10^6^ flight fatigue loads. However, there were obvious fluctuations in the load-displacement curves, which may have been caused by debonding between the stiffeners and the skin. Experimental results and the simulation matrix show that the post-buckling ratio increased with the increase of the stiffness ratio, then was stable after 2.0. Furthermore, the thinner the skin, the greater the post-buckling ratio. The experimental and simulation results provide an important reference for the structural design and failure-mechanism analysis of composite stiffened panels.

## 1. Introduction

Composite materials, especially carbon-fiber-reinforced resin matrix composites, can demonstrate excellent properties including high specific strength, high specific modulus, strong designability, and good durability resistance [1,2,3]. Due to their high flexural stiffness and low structural weight [4], composite stiffened panels are widely used in the aerospace industry, such as aircraft fuselage, radomes, wings, etc., significantly improving the efficiency of typical load-bearing structures [5]. However, epoxy resin matrix composites are sensitive to impact damage because of low fracture strength in the thickness direction and their obvious brittleness. In the process of aircraft manufacturing, maintenance, and daily use, stiffened panels are inevitably subjected to low-speed impacts, such as by collision or falling tools, resulting in visible impact damage (VID) or barely invisible impact damage (BVID) on the structural surface. Inside the structure, matrix cracking, interlaminar delamination, and even local fiber fractures can only be detected using nondestructive testing (NDT). These types of internal damage that are difficult to locate can seriously degrade the mechanical properties of composite stiffened panels and pose a potential threat. Therefore, it is necessary to estimate impact damage and the residual strength of composite stiffened panels after low-speed impact, which will be helpful to the potential application of composites [6,7,8,9].

Using the Mindlin plate theory and a nine-node Lagrangian element, Tiberkak [10] studied the dynamic response of fiber-reinforced composite laminates under low-speed impact. The analysis showed that the contact force increased with an increase in the contact ratio to 90°, and transverse shear stress was dominant. Based on the continuous damage mechanics (CDM) method, Wei Sun [11] used the softened inclusion model combined with an analysis of material performance reduction to describe impact damage sustained by stiffened panels. It was concluded that impact damage reduced the failure load of stiffened panels to varying degrees. When the impact energy was 80 J, the failure load decreased by 44% at most, and when impact energy was between 34 J and 44 J, the failure load was more sensitive to impact energy. Riming Tan [12] found that the impact energy of 23 J caused serious impact damage to single T-stiffened panels, including fiber fracture, delamination, and debonding of the stiffener–skin interface. Impact damage promoted buckling of the stiffened panels under the compressive load, and reduced the buckling load and failure load by 9.62% and 34.1%, respectively. Soto. A [13] applied analysis of continuous damage mechanics and cohesive elements to establish intralaminar and interlaminar models, predicted the responses of large composite stiffened panels to low-speed impact and compression after impact, and discussed the effects of parameters including penalty stiffness and element size on the results of the numerical model. Wu Xiaoyang [14] and Wu Qianqian [15] investigated the buckling and post-buckling loads of composite stiffened panels under multi-point impact damage experimentally and by simulation. Within the existing research on impact and compression after impact in composite stiffened panels, most scholars have focused on the damage modes caused by impact, and the effects of impact damage on compression. There is a lack of mature analysis describing the modeling of the impact area and the integrated simulation of impact and compression after the impact [16,17,18,19]. In addition, due to the high costs of manufacturing and testing composite stiffened panels, research has been scarce on the post-buckling history of stiffness in stiffeners and skins. Wang Yuequan [20] experimentally tested 18 specimens and investigated the stiffness and buckling and post-buckling behavior of stiffened composite panels. 

When the damage caused by fatigue load reaches the allowable limit for composite stiffened panels, structures lose their bearing capacity. Fatigue damage cannot be ignored in structural design, so it is of great significance to study the influence of fatigue loading on the post-buckling behavior of composite stiffened panels. Feng Yu [21] studied the effect of fatigue load on the buckling and post-buckling behavior of composite stiffened plates under CAI testing. The results showed that fatigue load had no obvious effect on impact damage, buckling load, or buckling mode, but fatigue load reduced the ultimate bearing capacity by about 6.1%. Raimondo. A [22] and others studied the propagation of delamination in composite structures under fatigue load based on virtual crack-closure technology (VCCT), and analyzed double cantilever beams and mixed-mode bending specimens. It has been demonstrated that the numerical method can predict the development of fatigue-associated delamination in single longitudinal beams after buckling. Guan Zhidong [18] studied the fatigue behavior and failure mechanism of composite stiffened panels under shear fatigue experiments. Overall, there have been few studies on the effect of fatigue load on the buckling and post-buckling behaviors of composite stiffened panels, and a detailed description of the relevant processes has not been achieved [23,24,25,26]. 

In this paper, four different types of composite stiffened panels were carried out on low-velocity impact, a secondary fatigue test, and a CAI test. A two-dimensional Hashin failure criterion was written into a user-defined field subroutine (USDFLD) to judge the intralaminar failure of composite stiffened panels. The impact damage was introduced into the process of compression by defining a predefined field. Through the comparison of the test and simulation results, the simulation model with high accuracy has been certified, which can be used to predict the buckling load and the post-buckling load of stiffened composite panels in engineering. In addition, the influence of fatigue damage on residual strength after impact on composite stiffened panels was also studied experimentally. Based on the experimental results and the simulation matrix, the relationship between the post-buckling ratio and the skin/stringer stiffness ratio was analyzed. The experimental and simulation results provide an important reference for the structure design and failure mechanism analysis of composite stiffened panels.

## 2. Experimental Set-Up

### 2.1. Specimen and Material Properties

A carbon-fiber-reinforced composite stiffened panel was made of a flat skin and four I-shaped stringers, as shown in Figure 1. The skin and stringers were manufactured and co-cured as an integral structure using an autoclave. The four sets of specimens with different skin and stringer are denoted by C1, C2, C3, and C4 respectively. The material of C1 and C2 is T300/BA9916, and that of C3 and C4 is T300/5228A. Firstly all the specimens were subjected to low-velocity impact, and then half of them were directly subjected to a compression test, and half of the others were subjected to fatigue tests before being compressed. The purpose of the fatigue test before being compressed is to obtain the residual fatigue life of the aircraft after low-speed impact. If the specimen has not been damaged after specified fatigue times, check whether the residual strength of the structure is still within the design range or not. Therefore, the secondary fatigue damage is considered the mechanical behavior of composite stiffened panels in the full life cycle. Details of the specimens are listed in Table 1. The fixed and loading ends with 50 mm long were fixed in Aluminium metallic boxes filled with epoxy resin. The stacking sequence of skins and stringers is given in Table 2. The material properties are shown in Table 3.

### 2.2. CAI Test

The low-velocity impact tests of specimens were carried out on a drop-weight impact testing machine. The required impact energy of 50 J was obtained by adjusting the mass and the height of the impactor. The impactor with a diameter of 12.7 mm and a mass of 10 kg is a smooth hemispherical steel striker. The specimen was clamped, the stiffened side placed down, and the impact position located at the edge of the second stringer in the longitudinal center of the specimen, as shown in Figure 2. After the punch hit the specimen, the secondary impact prevention device was activated to hold the impactor, and thus the specimen was only impacted once. The specimen after the low-velocity impact was detected by NDT, and the damaged area was marked on the specimens. The specimens with impacted damage were then used in the compression tests and fatigue tests. 

The compression tests were conducted on a hydraulic fatigue testing machine with a 5000 kN load sensor. The compression test fixture is a load guide device composed of four guide pillars, a top loading plate, and a base plate. The top loading plate can only move vertically downward along the guide pillars, and the axial load was applied to the neutral Axis of the specimens during loading. The loading and fixed ends of the specimens were clamped, and two sides of the specimens were simply-supported by side plates, as shown in Figure 3a. The photo of the compression test fixture is shown in Figure 3b. In order to capture the buckling load, many strain gauges were bonded back-to-back on the skins and stringers. The location of strain gauges is shown in Figure 4. The displacement controlled loading method was performed with a velocity equal to 0.5 mm/min. The displacement was recorded as the movement of the machine’s crossbeam, and the load value was measured by the machine load cell. Before the experiment, the trial test would be carried out to make sure the stiffened panels are uniformly loaded, and the basis is to ensure that the difference of back-to-back strain gauges value is less than 5% before 30% of the ultimate load. If the requirement is not met, it is necessary to finely adjust the front and rear positions of the fixed end and loading end with a metal sheet of appropriate thickness.

### 2.3. Fatigue Test

The test machine and fixture used in the fatigue test are the same as those used in the compression test. The constant amplitude compressive-compressive fatigue load is studied, and the stress ratio R is 10. According to the aircraft design requirements, the number of fatigue cycles is 10^6^. The maximum fatigue loads of four sets of specimens are 151 kN, 309 kN, 77.5 kN, and 427 kN respectively, which are about 20~40% of the respective buckling load and 10~32% of the failure load. During the fatigue test, NDTs were conducted to monitor the propagation of the impact damage. Before 8 × 10^5^ cycles, NDT was carried out every 2 × 10^5^ cycles, and then every 10^5^ cycles. After 10^6^ cycles of fatigue test, the specimens were then subjected to compression test.

## 3. Failure Criterion and Damage Evolution

The damage of composite structures usually involves different failure modes: fiber failure, matrix failure, fiber matrix shear failure, and delamination. Through NDT, it is found that the damage mainly occurs on the skins or stringers, and there is no debonding between the skins and stringers. It is because the interface between the skin and the stiffeners is stronger than that between the skin or stiffener ply. In addition, the tests show that the damage between composite laminates under low-speed impact is mainly matrix damage, which can cover the damaged area of delamination. Therefore, this paper adopts a two-dimensional shell model to analyze [26,27] the failure of composite stiffened panels under low-velocity impact and CAI. 

### 3.1. Failure Criteria and Degradation Procedure

In this paper, the Hashin criterion is used to predict the intralaminar damage of stiffened panels, including five failure modes: matrix tension, matrix compression, fiber tension, fiber compression, and fiber-matrix shear failure. When the corresponding elements meet the Hashin criterion, the field variables and state variables are defined as 1.0, and the stiffness reduction criterion is used to reduce the stiffness of damaged elements. When damaged elements reach a certain degree, the structure lost its bearing capacity, and damage initiation, propagation, and failure of composites are simulated [6,28,29]. The expressions of Hashin criterion are shown in Table 4. Wherein, $\sigma_{1}$, $\sigma_{2}$ and $\varepsilon_{12}$ represent normal stress and shear stress in principal axis coordinate system. These parameters are obtained through material mechanical property tests of unidirectional plates [30].

With the increase of compression load, the corresponding integration points begin to fail according to the failure criterion, and the element will lose its bearing capacity, which is mainly reflected in the reduction of the stiffness. The original load born by the failure element will be transmitted to the surrounding elements, and the size of the transferred load is determined by the residual stiffness of the damaged element. The transfer of force and load also affects the final bearing capacity of the structure. Therefore, it is necessary to reduce the stiffness of the integral point of the material in the finite element simulation. Generally speaking, the elastic modulus $E_{f}$ after reduction can be expressed as Formula (1):(1)$$E_{f}=d\cdot E_{0}$$

$E_{0}$ is the initial elastic modulus, *d* ranging from 0 to 1 is the reduction factor which is related to different failure modes. The reduction factor is closely related to the material type, interface bonding degree, stress state, and other factors, and it can not be accurately acquired through the existing experimental methods. The stiffness reduction criterion proposed by Tan [31] is used in this paper, as shown in Table 5.

### 3.2. Finite Element Modelling

In this paper, Finite element models of impact and CAI test of composite stiffened panels were established by ABAQUS/Explicit module, and the user subroutine was compiled to judge the damage initiation and failure mode of impact and CAI. 

The CAI process is actually static loading, but the overall stiffness becomes singular due to the weakening of element stiffness with compression load increases. Dynamic module which adopts explicit difference calculation method solving the non-convergence problem is generally used in damage generation and propagation simulation. Quasi-static loading is achieved by setting a long calculation time, such as 1.0 s even longer, and calculation efficiency can be improved by setting a mass scale factor. 

The contact between the impactor and the skin was defined by the contact algorithm based on the penalty stiffness. During the process of impact, four edge sides of stiffened panels were fixed, and the impactor directly hit the specimen with an initial speed of 3.16 m/s. During the process of compression, the displacement load was applied on the loading end. All the translation degrees and rotation degrees were limited on the fixed end, and symmetry constraint in the thickness direction was added on the other three edges. The S4R element with high accuracy was adopted in the FEA model. The mesh size is about 5 mm, and the total number of grids is 70,530. The FEM model and boundary condition are shown in Figure 5.

Figure 6 gives the flow chart of strain and damage calculation of the stiffened panels under impact and CAI. In the first step, the impact analysis is carried out, and the calculation time must be long enough to ensure that the vibration of the plate after impact has been dissipated, and then the results including deformation and damage are taken as the initial state of the second step. In this way, the element damage and structural deformation are introduced into the compression loading process of composite stiffened panels. 

## 4. Impact and CAI Experiment and Simulation Results

### 4.1. Impact Test Result

NDT was conducted after the impact test, and the damaged areas are recorded with a white line in Table 6. The shape of impact damage is approximate to the ellipse, and the long axis is along the direction of stiffeners, and the short axis is perpendicular to stiffeners. The “a” represents the length of the long axis, and the “b” represents the length of short axis. The damaged area is defined as a × b. Under the impact energy of 50 J, the long axis value is between 70~100 mm and the short axis is between 35~41 mm. The size of the damaged area is related to many factors, such as the cross-sectional size of stiffeners, the spacing of the stiffeners, the material, and the thickness of the skin. Comparing four sets of composite stiffened panels, it can be seen that the ply ratio and the thickness of the skin are the main factors. For example, as the skin thickness of C2 and C4 are the same, the long axis of damage is close to each other. Compared with C1, C2, and C3, the long axis decreases with the increase in skin thickness, but the relationship is not absolutely linear. It can be found that the short axis is strongly related to the ply ratio, the increase of 0° will reduce the value, and the ply ratio of 90° and ±45° has little effect. 

The damaged shape of C1 is shown in Figure 7, and the simulation result is compared with the experiment. The red area represents the elements damaged, and the blue area represents the undamaged elements. The damage simulated includes skin damage and stiffener damage, and the sum of the two is the total damage. The damaged area is about 85 mm× 45 mm, which is a little bigger than the test average value of 79.8 × 41 mm^2^, and the long axis by simulation is 6 mm longer than the test result. As the impact point is at the center of the stiffener edge, the damage is along the direction of the stiffener, and the damaged area near the free end is larger. It is mainly caused by weak support near the free end. 

### 4.2. Influence Analysis after Introducing Fatigue Damage

Back-to-back strains measured on the transverse middle of the skin are selected to analyze buckling and post-buckling behavior. Figure 8 shows the strain-load curves of composite stiffened panels without fatigue test and with fatigue test. Back-to-back strains measured on the stiffener web are selected, and the strain-load curves are shown in Figure 9. Comparing C1-static with C1-fatigue, the trends of the strain-load curve are basically consistent. Fatigue load has little effect on the strain before buckling, but the strain load curves with fatigue damage are not as smooth as that without fatigue damage. This is because the specimens have some damage under fatigue load.

At the beginning of loading, the strain values on the two sides of the skin are almost the same. When the load reaches a certain value, the strains begin to bifurcate. It can be judged that the skin has buckled at the first bifurcation position. Local buckling first occurred in the middle of the stiffened panel, that is, bulges appeared in the transverse direction, and the concave and convex alternately appeared. With the increase in load, the bulges gradually appeared in the area near the loading end and the fixed end. The bending strains become bigger and bigger, and the buckling shape of the stiffened panel becomes more and more prominent.

The strain-load curves on the stringers are absolutely linear before local buckling occurs on the skin. After local buckling of the skin, the compressive load is redistributed, the stiffeners bear more load, and the strain curve on the stiffeners has an obvious turning point. With the increase in load, the change rate of strain increases. Before the failure of the composite stiffened panel, the strain on the stiffeners has not forked, indicating that the stiffeners have been in the main bearing state. 

The buckling load and failure load of all the specimens are summarized in Table 7. The post-buckling ratio is defined as the post-buckling load divided by buckling load, which is used to study the bearing capacity after skin local buckling of different stiffened panel structures under compression load. The post-buckling ratio of C1, C2, and C4 is about 1.3 to 1.4, while that of C3 is about 2.1. This is because the skin of C3 is relatively thinner and prone to buckle locally, and then the load is transferred to the stiffeners which bear a large load. Therefore, the post-buckling ratio of C3 is longer than that of the other sets, and thin skin can give full play to the potential of the structure. The buckling load and failure load are related to many factors, such as ply ratio, the cross-sectional size of stiffeners, the spacing length of stiffeners, etc. Comparing the results of C1 and C2, it can be concluded that the increase in the number of 0° skin layers will enhance the bearing capacity of stiffened panels and increase the value of buckling load and failure load. 

The load-displacement curves for all the specimens are drawn in Figure 10. The displacement in the figure is the testing machine, which includes the clearance of the testing machine, and the elastic deformation of the testing machine structure, etc. In order to analyse the difference between the test curve and the simulation curve, zero clearing of displacement is not used. From Table 7 and Figure 10, the influence of the fatigue test on buckling load and failure load for composite stiffened panels can be concluded. Fatigue load has little effect on the buckling load and failure load. For C1 and C2, the fatigue test reduces the buckling load by 3.8%, 9.7%, and 4.9%, and 5.9% for failure load. Yet for C3 and C4, the buckling load and failure load for the specimens with fatigue damage are almost the same as that without fatigue damage. It is abnormal to obtain such a result, and many factors can affect it, such as materials, processes, etc. This result shows that fatigue tests have little influence on the buckling and post-buckling load of composite structures, the reduced ratio is about 5%.

From Figure 10, it can be seen that the largest difference between the curves with fatigue damage and that without fatigue damage is whether there are one or more turning points or not. There are one or more turning points on the load-displacement curve with fatigue damage during the loading process, but the load-displacement curve of the specimens without fatigue damage is relatively smooth. The turning points are marked on the load-displacement curves. The reason for the fluctuation of the load-displacement curve is caused by local buckling or local failure. Through the analysis of the strains, it is found that there is no local buckling and failure at the turning point, and it is considered that the fluctuation is caused by local debonding between the stiffeners and the skin. Since NDT is not conducted during compression loading, there is no test basis, and the influence of the relevant adhesive layer on fatigue can be further developed in the subsequent research. The fatigue load degrades the adhesive film and then weakens the buckling and failure load of the composite stiffened panels.

Fatigue load is the design load of the aircraft structure, and the fatigue cycle is 10^6^, which meets the requirements of aircraft design. The experiment result shows that fatigue load has little effect on the buckling and post-buckling of composite stiffened panels. However, attention should be paid to the weakening of adhesive strength, and the results can be of great significance for the design of aircraft structures.

### 4.3. Failure Damage Propagation of CAI

The buckling load of the composite stiffened panel was calculated by the eigenvalue analysis method, and the failure load was obtained by dynamic analysis using the Hashin failure criterion and Tan’s stiffness reduction. The buckling load and failure load obtained by the FEA method are marked in Figure 10. It can be seen that the simulation result and experiment result are almost the same, and the errors of the buckling load and failure load are less than 10%. It can be concluded that the buckling and post-buckling behavior of composite stiffened panels can be predicted by the FEA model.

Figure 11 gives the fracture photos and load-displacement curves of composite stiffened panel C1 and C3. From Figure 11, it can be seen that the simulation curves and the test curves have the same slope and turning point. The load-displacement curves in the middle section are relatively good. In the initial stage of loading, there are problems such as fixture clearance and internal prestress of the specimens, and the simulated load-displacement curves are different from that of the test. After local buckling, some elements fail, the nonlinearity of the structure is very large, and the simulation result has a certain error. In general, the curve can well simulate the loading process. The buckling first occurs on the skin, that is, uneven bulges appear, and the position is close to the impact area. With the increase of the load, the waveform gradually changes from first-order to second-order. Subsequently, the bulge is larger and larger, and the buckling mode is gradually stable. When the equivalent stress at wave crest and wave trough meets the failure criterion, the elements fail, the load transfers to the stiffeners, and finally, the whole stiffened panel loses its bearing capacity. The failure of the stiffened panel is mainly caused by the excessive deformation at the wave crest and wave trough, which leads to the fracture of the skin and stiffener. The damages appear at the impact position and then expand to both sides, perpendicular to the loading direction, and finally form a horizontal fracture in the middle of the stiffened panels, which are basically consistent with the fracture mode and failure location of the test.

During the fatigue test, NDT results show that the damage did not expand under the fatigue load specified in this paper. After the fatigue test, the structural rigidity remained almost unchanged, so the fracture mode of CAI with fatigue damage is consistent with the specimen without fatigue damage.

Figure 12 illustrates the damage generation and propagation of panel C1 under compression load. MC, MT, FC, FT, and FMS represent matrix compression failure, matrix tensile failure, fiber compression failure, fiber tensile failure, and fiber-matrix shear failure, respectively. The red color indicates that the element is completely damaged and loses its bearing capacity. 

According to Figure 12, the initial damage is matrix tensile and occurs at the impact area, which is caused by low-velocity impact. Then, fiber compression, matrix compression, and fiber-matrix shear failure occur. When the load is 716 kN, the area of matrix compression, fiber compression, and fiber-matrix shear failure is significantly enlarged. When the load is 986 kN, the stiffened panel is completely destroyed, and the failure mode of CAI is mainly fiber compression, matrix tensile, and fiber-matrix shear failure. CAI damage simulation shows that damage begins at the impact position, then expands from the middle to both sides perpendicular to the loading direction, and finally forms a relatively flush transverse fracture in the middle of stiffened panels. From the above conclusion, it is concluded that the Hashin criterion can well predict the buckling and post-buckling behavior of composite structures. 

## 5. Influence of Skin/Stringer Stiffness Ratio on Post Buckling Process

In order to study the influence of skin/stiffeners stiffness ratios on post-buckling, 15 stiffened composite panels with different skin/stiffener stiffness ratios were carried out to study the buckling and post-buckling behaviors.

The skin/stringer stiffness ratio is defined as
(2)$$\lambda =\frac{{\left(EA\right)}_{skin}}{n{\left(EA\right)}_{stiffeners}}$$

${\left(EA\right)}_{skin}$ is the equivalent stiffness in the axial direction of the skin, which is the equivalent modulus of the skin multiplied by the cross-sectional area of the whole composite panel. ${\left(EA\right)}_{stiffeners}$ is the axial equivalent stiffness of a single stiffener, and *n* is the number of stiffeners on a stiffened panel.

Table 8 gives the ply information of skins and stringers. Additionally, the results of the buckling and post-buckling of 15 panels are listed in Table 9. From Table 9, it can be seen that buckling load has a great relationship with the stiffness of the stiffeners, and buckling load increases with the increase in the stiffeners stiffness. Comparing stiffeners with 16 plies or 32 plies for the same skin, it is found that when the stiffness of the stiffeners is doubled, the buckling load is nearly doubled. This conclusion is consistent with the trend of the empirical formula, which shows that the boundary condition has a great influence on the results when the local buckling of skin is calculated.

Here, Figure 13 shows the relationship between the post-buckling ratio and the skin/stiffener stiffness ratio. From Figure 13, it can be seen that when the skin ply is the same, the post-buckling ratio increases with the increase of the skin/stiffeners stiffness ratio. The stiffener stiffness not only affects the local buckling load, but also bears the load transferred by the skin after local buckling, and the stiffeners stiffness directly determines the bearing capacity of the whole panel structure. However, post-buckling ratio does not always increase with the increase of the skin/stiffeners stiffness. The change of post-buckling ratio is not obvious after a certain proportion. No matter thin skin or thick skin, when the skin/stiffener stiffness ratio reaches 2.0, post-buckling process will not grow too fast. Considering the additional weight by stiffener thickness, it is not worth it. Therefore, it is recommended that the skin/stringer stiffness ratio should be less than 2.0 in the composite stiffened panels. Of course, the thickness of the skin and the shape of the stiffeners have a great relationship with the structural load. Therefore, it is necessary to optimize the design of stiffened panels.

Comparing the post-buckling ratio for different skin layers S1, S2, and S3, it can be found that the thinner the skin, the more obvious the post-buckling ratio is. For the thick skin, the load is mainly borne by the skin itself, and the load transferred to the stiffeners after the local buckling is already very large, which almost exceeds the bearing capacity of the stringer. So the stiffeners damage quickly, and the post-buckling ratio is very small. Yet for the thin skin stiffener panel, the skin is easy to buckle, and the load is transferred to the stiffeners after the local buckling is gradually increased. The nonlinear characteristic of the structure is very strong, and the post-buckling ratio is bigger.

## 6. Conclusions

The impact and CAI behaviors of four sets of composite stiffened panels were investigated through experiments and simulations. In addition, the influence of flight fatigue load on the residual strength of composite stiffened panels was analyzed. Through systematic research and analysis, some useful conclusions can be drawn.

(1)The shape of impact damage is approximate to the ellipse, the long axis is along the direction of the stiffeners, and the short axis is perpendicular to stiffeners. The size of the damage area is related to many factors, such as the cross-sectional size of stiffeners, spacing of stiffeners, the material and the thickness of the skin. Comparing four sets of stiffened panels, it is found that the ply ratio and the thickness of the skin are the main factors.(2)The influence of fatigue damage on residual strength after impact was also studied experimentally, with results showing that the buckling and failure loads decreased by about 5% under 10^6^ flight fatigue loads. However, there are obvious fluctuations in the load-displacement curves, which may have been caused by debonding between the stiffeners and the skin.(3)The buckling load, failure load, and failure mode obtained from the simulation model established in this paper are in good agreement with the tests, so the 2D shell model combined with the Hashin criterion and stiffness degradation can meet the need for evaluation of composite stiffened panels in engineering.(4)Based on the experiment results and the simulation matrix, the relationship between post-buckling ratio and skin/stringer stiffness ratio is analyzed. The results show that the post-buckling ratio increases with the increase of the skin/stiffness ratio, and is then stable after 2.0. Furthermore, the thinner the skin, the greater the post-buckling ratio.

## Figures and Tables

**Figure 1 polymers-15-00165-f001:**
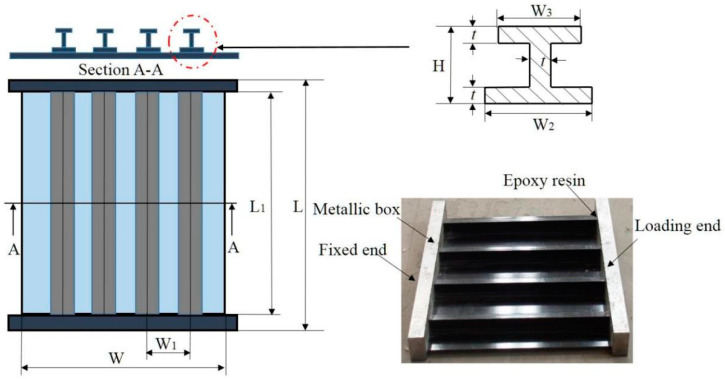
Schematic diagram and physical drawing of the specimen.

**Figure 2 polymers-15-00165-f002:**
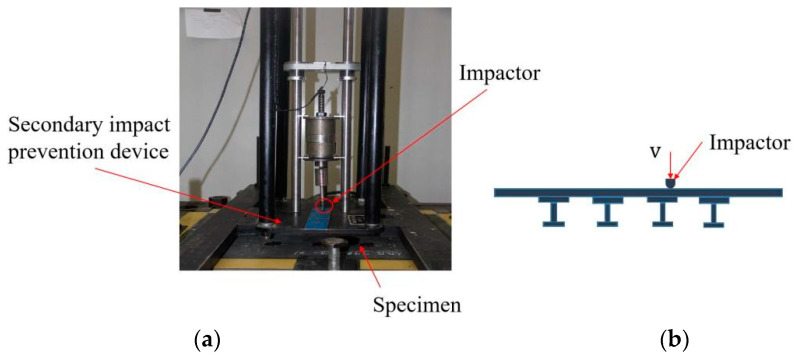
The impact test device and the schematic diagram of impact position. (**a**) Impact test device; (**b**) Impact position.

**Figure 3 polymers-15-00165-f003:**
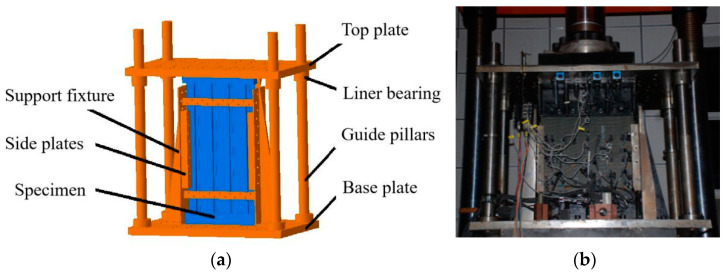
CAI test fixture and installation diagram. (**a**) Schematic diagram of compression test fixture; (**b**) Photo of compression test.

**Figure 4 polymers-15-00165-f004:**
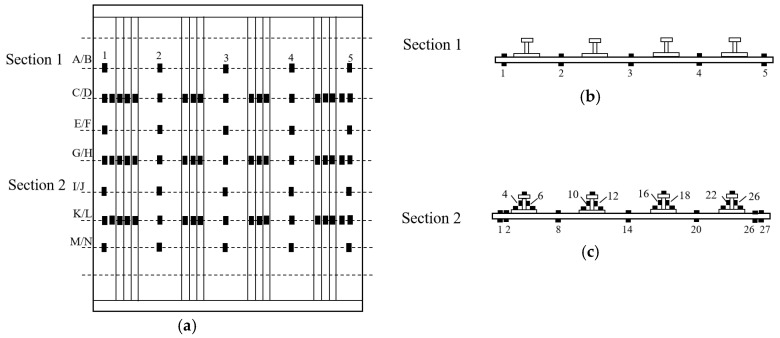
Location of strain gauges. (**a**) Distribution map; (**b**) Cross A/B, E/F, I/J and M/N; (**c**) Cross C/D, G/H and K/L.

**Figure 5 polymers-15-00165-f005:**
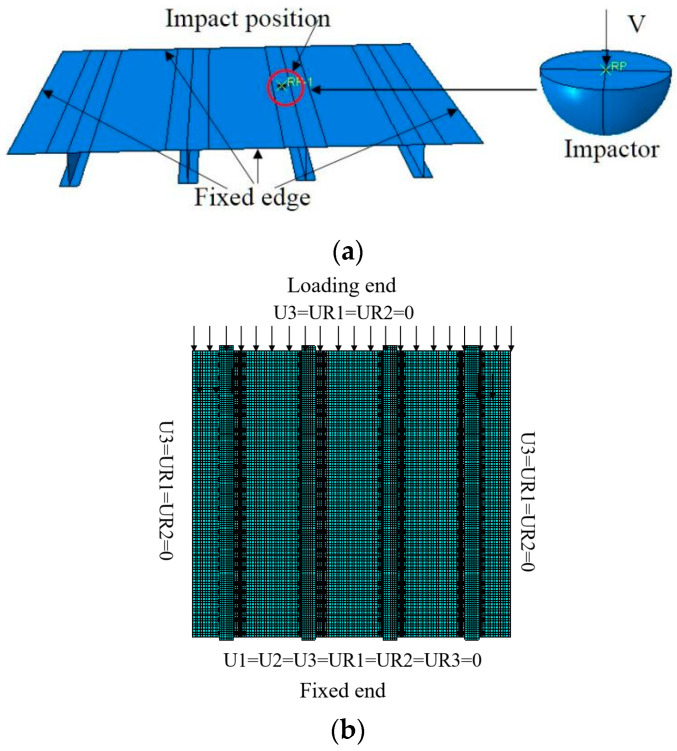
Finite element model and boundary condition. (**a**) Impact simulation; (**b**) Compression simulation.

**Figure 6 polymers-15-00165-f006:**
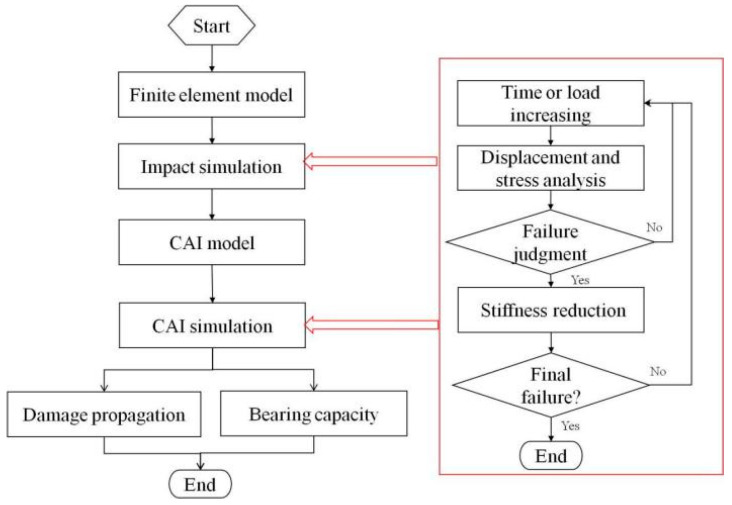
Flow chart of finite element analysis.

**Figure 7 polymers-15-00165-f007:**
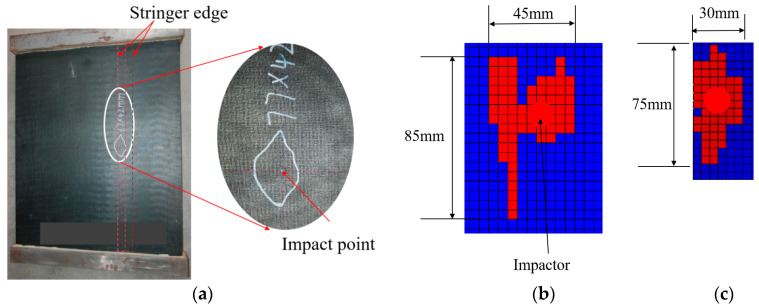
Impact position and shape: (**a**) Impact position and shape, (**b**) Damage on the skin, (**c**) Damage on the stringer.

**Figure 8 polymers-15-00165-f008:**
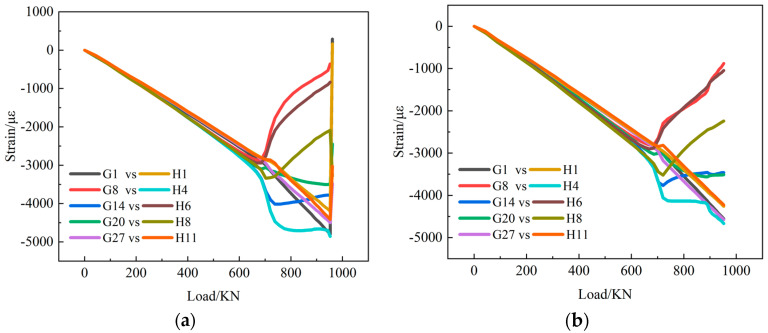
Back-to-back strain measuring points on the skin. (**a**) C1-Static-1; (**b**) C1-Fatigue-1.

**Figure 9 polymers-15-00165-f009:**
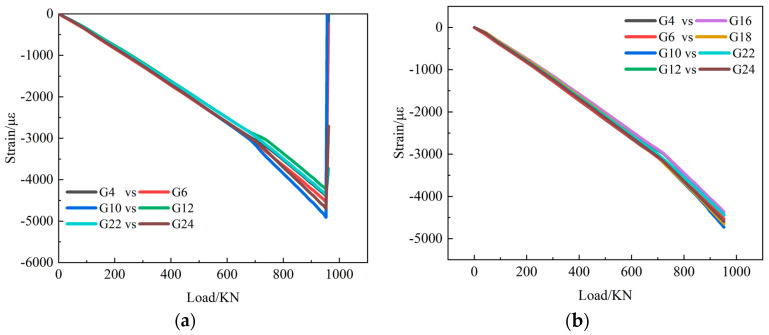
Back-to-back strain measuring points on stiffener web. (**a**) C1-Satic-1; (**b**) C1-Fatigue-1.

**Figure 10 polymers-15-00165-f010:**
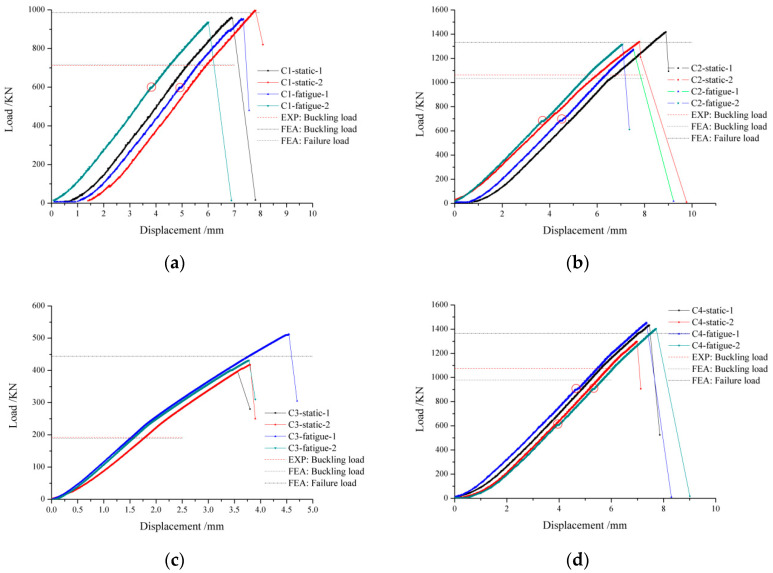
The load-displacement curves for all the specimens. (**a**) C1; (**b**) C2; (**c**) C3; (**d**) C4.

**Figure 11 polymers-15-00165-f011:**
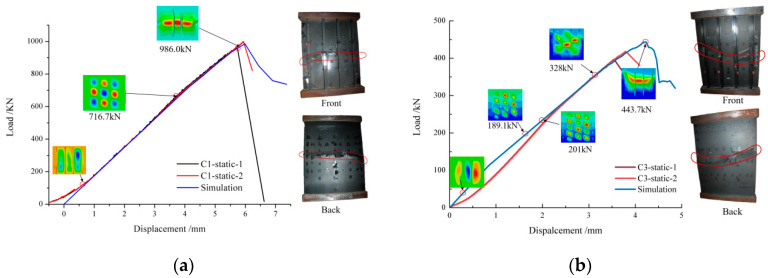
Fracture photos of specimen and load-displacement curves: (**a**) C1, (**b**) C3.

**Figure 12 polymers-15-00165-f012:**
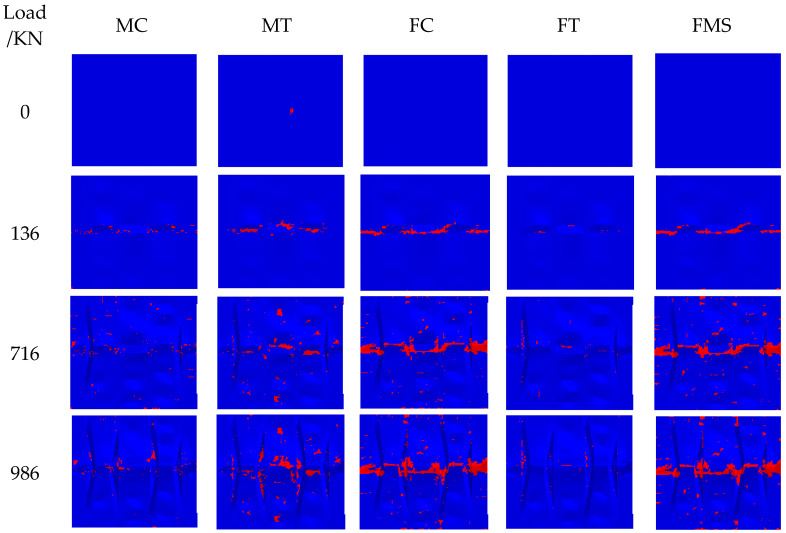
Damage generation and expansion of Panel C1.

**Figure 13 polymers-15-00165-f013:**
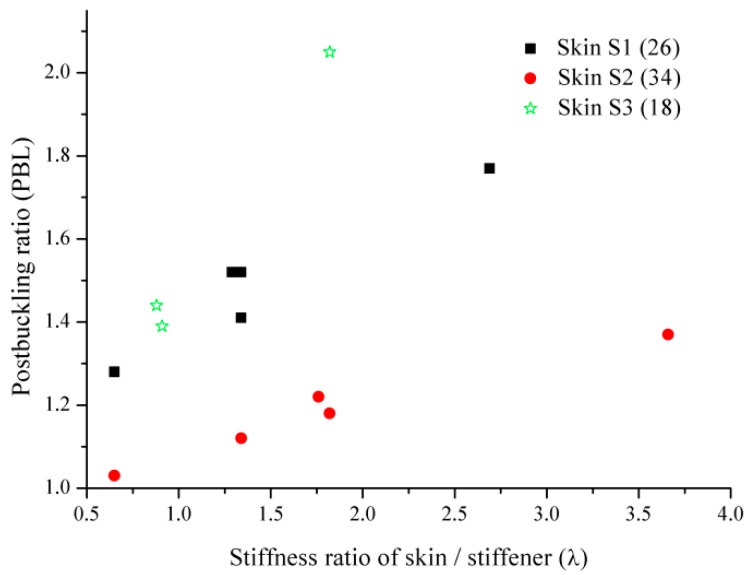
The relationship between post-buckling ratio and skin/stringer stiffness ratio.

**Table 1 polymers-15-00165-t001:** The dimensions of specimens (unit: mm).

Code	Parameter	Four Types of Composite Stiffened Panel			
		C1	C2	C3	C4
*L*	Total length	650	670	650	605
*W*	Width	612	612	600	600
*L* _1_	Effective length	550	570	550	505
*W* _1_	Rib spacing	153	153	150	150
*H*	Rib Height	45	45	30	40
*W* _2_	Width of Rib inner edge	55	55	50	50
*W* _3_	Width of Rib outer edge	25	25	20	20
*t*	Rib thickness	3.25	3.25	2	3.5

**Table 2 polymers-15-00165-t002:** The stacking sequence of the skin and stringer.

Set	Skin	Stringer
C1	[−45/0/−45/0/45/45/0/45/90/0/90/−45/0]_S_	[−45/0_2_/45_2_/0_2_/−45/0_2_/45/90/−45]_S_
C2	[−45/0/−45/0/45_2_/0_2_/45/90/−45/0/90/−45/0_2_/45]_S_	[−45/0_2_/45_2_/0_2_/−45/0_2_/45/90/−45]_S_
C3	[45/45/0_3_/−45/90/0/90]_S_	[0/45/−45/90/45/0_2_/−45]_S_
C4	[45/45/0_3_/−45/90/45/0_3_/−45/90/0_4_]_S_	[0_2_/45/0_2_/−45_2_/90/45_2_/0_3_/−45]_S_

**Table 3 polymers-15-00165-t003:** Material properties of the unidirectional lamina.

Physical	Code	T300/BA9916	T300/5228A
Longitudinal Young modulus	$E_{1}$ (GPa)	135	127
Transverse Young modulus	$E_{2}$ (GPa)	9.63	8.7
Poisson’s ratio	μ	0.3	0.32
In-plane shear modulus	$G_{12}$ (GPa)	4.69	3.8
Longitudinal tensile strength	$X_{T}$ (MPa)	1805	3150
Longitudinal compressive strength	$X_{C}$ (MPa)	1338	1494
Transverse tensile strength	$Y_{T}$ (MPa)	55.1	94.2
Transverse compressive strength	$Y_{C}$ (MPa)	204	50
In-plane shear strength	*S* (MPa)	130	230
Ply thickness	T (mm)	0.125	0.125
Density	ρ (g/cm^3^)	1.6	1.6

**Table 4 polymers-15-00165-t004:** Hashin criterion.

Failure Mode	Failure Criterion
Matrix compression failure	${\left(\frac{\sigma_{2}}{Y_{c}}\right)}^{2}+\frac{2\tau_{12}{}^{2}/G_{12}+3\alpha \tau_{12}{}^{4}}{2S^{2}/G_{12}+3\alpha S^{4}}\geq 1$
Matrix tensile failure	${\left(\frac{\sigma_{2}}{Y_{t}}\right)}^{2}+\frac{2\tau_{12}{}^{2}/G_{12}+3\alpha \tau_{12}{}^{4}}{2S^{2}/G_{12}+3\alpha S^{4}}\geq 1$
Fiber compression failure	$-\frac{\sigma_{1}}{X_{c}}\geq 1$
Fiber tensile failure	${\left(\frac{\sigma_{1}}{X_{t}}\right)}^{2}+\frac{2\tau_{12}{}^{2}/G_{12}+3\alpha \tau_{12}{}^{4}}{2S^{2}/G_{12}+3\alpha S^{4}}\geq 1$
Fiber matrix shear failure	${\left(\frac{\sigma_{1}}{X_{c}}\right)}^{2}+\frac{2\tau_{12}{}^{2}/G_{12}+3\alpha \tau_{12}{}^{4}}{2S^{2}/G_{12}+3\alpha S^{4}}\geq 1$

**Table 5 polymers-15-00165-t005:** Stiffness reduction criterion.

Failure Mode	Stiffness Reduction Criterion
Matrix compression failure	$E_{f}=0.4E_{0}$(Q=$E_{22}$, $G_{12}$, $\upsilon_{12}$)
Matrix tensile failure	$E_{f}=0.2E_{0}$(Q=$E_{22}$, $G_{12}$, $\upsilon_{12}$)
Fiber compression failure	$E_{f}=0.14E_{0}$(Q=$E_{11}$, $G_{12}$, $\upsilon_{12}$)
Fiber tensile failure	$E_{f}=0.07E_{0}$(Q=$E_{11}$, $G_{12}$, $\upsilon_{12}$)
Fiber matrix shear failure	$E_{f}=0.2E_{0}$(Q=$G_{12}$, $\upsilon_{12}$)

**Table 6 polymers-15-00165-t006:** Impact damage photos and area.

Type		C1	C2	C3	C4
Damage photos		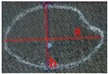	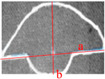	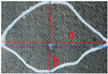	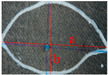
Damage area/mm (Four repeat specimens)		65 × 40 95 × 41 77 × 42 82 × 41	66 × 36 65 × 42 100 × 39 64 × 40	82 × 35 112 × 38.5 79 × 41.5 115 × 45.6	82 × 39 76 × 24 75 × 40 70 × 38
Average damage area/mm		79.8 × 41	73.8 × 39.3	97 × 40.2	75.8 × 35.3
Skin thickness/mm		3.25	4.25	2.25	4.25
Ply ratio	±45°	46.2%	47.1%	33.3%	29.4%
	0°	38.5%	41.2%	44.4%	58.8%
	90°	15.4%	11.8%	22.2%	11.8%

**Table 7 polymers-15-00165-t007:** Buckling load and failure load for all the specimens.

Panel Type	Identifier	BL/KN	FL/KN	PBL	Identifier	BL/KN	FL/KN	PBR
C1	C1-S-1	703	986.2	1.40	C1-F-1	703	953.1	1.36
	C1-S-2	721	998.6	1.38	C1-F-2	667	935.5	1.40
	FEA	712	986					
C2	C2-S-1	1075	1417.9	1.32	C2-F-1	920	1273.7	1.38
	C2-S-2	1050	1336.7	1.27	C2-F-2	994	1315.8	1.32
	FEA	1034.2	1332					
C3	C3-S-1	188	396.9	2.13	C3-F-1	208	512	2.27
	C3-S-2	195	418.0	2.14	C3-F-2	191	431	2.05
	FEA	189.0	443.7					
C4	C4-S-1	1105	1433.2	1.30	C4-F-1	1025	1450.1	1.41
	C4-S-2	1052	1297.7	1.23	C4-F-2	1010	1395.7	1.38
	FEA	977.8	1366.0					

NOTE: Buckling load is shorted as BL, failure load is shorted as FL, PBR is shorted as the post-buckling ratio.

**Table 8 polymers-15-00165-t008:** Ply information of skins and stringers.

Code	Ply	Total Plies	Equivalent Modulus/MPa	Area /mm^2^	EA /MPa.mm^2^
	Skin				
S1	[−45/0/−45/0/45/45/0/45/90/0/90/−45/0]_S_	26	68.252	1787.5	122,000.45
S2	[−45/0/−45/0/45_2_/0_2_/45/90/−45/0/90/−45/0_2_/45]_S_	34	71.128	2337.5	166,261.7
S3	[45/45/0_3_/−45/90/0/90]_S_	18	66.81	1237.5	82,677.38
	Stiffener				
J1	[0/45/−45/90/45/0_2_/−45]_S_	16	59.741	190.0	45,403.16
J2	[−45/0_2_/45_2_/0_2_/−45/0_2_/45/90/−45]_S_	26	76.351	308.8	94,293.485
J3	[0_2_/45/0_2_/−45_2_/90/45_2_/0_3_/−45]_S_	28	72.326	315.0	91,130.76
J4	[0/45/−45/90/45/0_2_/−45]_2S_	32	59.741	380.0	90,806.32
J5	[−45/0_2_/45_2_/0_2_/−45/0_2_/45/90/−45]_2S_	52	76.351	617.5	188,586.97

**Table 9 polymers-15-00165-t009:** Post-buckling ratio and skin/stringer stiffness ratio of simulation matrix.

Model	Skin Ply	Stiffener Ply	BL/KN	FL/KN	λ	PBR
Model-1	S1	J1	462.8	817.35	2.69	1.77
Model-2		J2	732.48	1116.67	1.29	1.52
Model-3		J3	827.04	1260.53	1.34	1.52
Model-4		J4	906.85	1279.71	1.34	1.41
Model-5		J5	1536.3	1965.6	0.65	1.28
Model-6	S2	J1	769.12	1051.47	3.66	1.37
Model-7		J2	1179.6	1441.49	1.76	1.22
Model-8		J3	1260	1482.58	1.82	1.18
Model-9		J4	1410.6	1471.98	1.34	1.12
Model-10		J5	1195.9	1434.82	0.65	1.03
Model-11	S3	J1	232.25	476.36	1.82	2.05
Model-12		J2	361.18	708.12	0.88	1.44
Model-12		J3	380.44	780.06	0.91	1.39

## Data Availability

The data supporting the findings of this manuscript are available from the corresponding authors upon reasonable request.

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
