# Peer review of "An Experimental and Numerical Study on Impact and Compression after Impact of Stiffened Composite Panels"

_polymers, 2022, doi:10.3390/polym15010165_

Round 1

Reviewer 1 Report

Experimental and Numerical study on the Impact and Compression after Impact of stiffened composite panels were conducted. Overall, the paper is well designed with rich experimental data and simulation results, it is significant to evaluate the impact performance of composites. The following comments are suggested to respond to make necessary improvement. 

1.     Abstract, please provide the research significance and background at the beginning. The agreement between the experimental results and numerical simulation should further be indicated. In addition, the failure mechanism analysis of composite stiffened panels is preferable.

2.     Introduction, the following comments and suggestions should be considered:

1)     In the first paragraph, composite panels should be further defined as fiber reinforced polymer composite. At the same time, its type, performance and advantages, such as light weight, high strength, good fatigue/durability resistances should also be further highlighted, especially for carbon fiber reinforced polymer composites related to this paper. Please review the following latest work for necessary supplement. Composite Structures 293, 115719. Composite Structures. 2021, 261: 113285. Polymer Composites 41 (12), 5143-5155.

2)     The authors have summarized the impact and fatigue damage of composite panels. However, the current summaries mainly focus on the analysis of properties evolution, and the mechanism analysis of impact damage and fatigue damage should be further considered to enrich this content.

3)     In the last paragraph of the introduction, the authors should emphasize the contribution, innovation and the significance for engineering applications of the current research work.

3.     For the test method, the authors claimed “Firstly all specimens were subjected to low-velocity impact, and then half of them were directly subjected to compression test, and the fatigue tests were applied to half of others before being compressed”, the specimens were subjected to loading of impact, compression and fatigue. Is there any practical engineering references for the above simulated loading working condition? Will the composite subject to the combined effects of the above three loadings in the service process?

4.     Why not conduct the micro-performance analysis and tests after the impact and fatigue tests, such as scanning electron microscope? This is very effective for revealing the damage and failure mechanism of materials, such as matrix cracking, fiber fracture, interface debonding and delamination damage, etc.

5.     The third part should focus on the finite element simulation considering the failure criterion and damage evolution. Therefore, the current subtitle may not be appropriate and needs further adjustment or modification.

6.     In figure 7, it is difficult to see the impact damage mode clearly. Do you have some local magnified pictures to clearly show the damage mode? In addition to Table 6, the results and related analysis on the impact strength and fracture energy are suggested.

7.     In table 7 and figure 10, the maximum buckling and failure load are more than 1000kN. On the other hand, the compression tests were conducted on an hydraulic fatigue testing machine with a 1000 kN load sensor. Whether there are some contradictions in these two parts, it is suggested to provide relevant explanations.

8.     In figure 11(a), for the load-displacement curves of C1, the initial load is 0 with the displacement. So it is recommended to further adjust the curves by moving to the left to analyze the relationship between some simulation and experimental results.

9.     Conclusions should be further improved including 3 to 4 key points.

Author Response

Response to Reviewer 1 Comments

Point 1: Abstract, please provide the research significance and background at the beginning. The agreement between the experimental results and numerical simulation should further be indicated. In addition, the failure mechanism analysis of composite stiffened panels is preferable. 

Response 1: A sentence has added to illustrate the importance, and the consistency between the experimental results and the numerical simulation has stated. At the end, the strength has replaced with failure mechanism. 

Point 2:Introduction, the following comments and suggestions should be considered:

1) In the first paragraph, composite panels should be further defined as fiber reinforced polymer composite. At the same time, its type, performance and advantages, such as light weight, high strength, good fatigue/durability resistances should also be further highlighted, especially for carbon fiber reinforced polymer composites related to this paper. Please review the following latest work for necessary supplement. Composite Structures 293, 115719. Composite Structures. 2021, 261: 113285. Polymer Composites 41 (12), 5143-5155.

2) The authors have summarized the impact and fatigue damage of composite panels. However, the current summaries mainly focus on the analysis of properties evolution, and the mechanism analysis of impact damage and fatigue damage should be further considered to enrich this content.

3) In the last paragraph of the introduction, the authors should emphasize the contribution, innovation and the significance for engineering applications of the current research work.

Response 2: 

1) The advantages of carbon fiber reinforced resin matrix composites have been supplemented in the introduction, and references have been cited.

2) This paper mainly studies the failure mode of low-speed impact and the influence of fatigue on the buckling and post buckling performance of stiffened panels. The effects of impact damage and the influence of fatigue are given in Part 2.

3) The last paragraph of the introduction has been completely modified.

Point 3:For the test method, the authors claimed “Firstly all specimens were subjected to low-velocity impact, and then half of them were directly subjected to compression test, and the fatigue tests were applied to half of others before being compressed”, the specimens were subjected to loading of impact, compression and fatigue. Is there any practical engineering references for the above simulated loading working condition? Will the composite subject to the combined effects of the above three loadings in the service process?

Response 3: This working condition simulates the actual loading condition, and the process is further described in Section 2.1. The purpose of fatigue test after impact is to obtain the residual fatigue life of aircraft after low-speed impact. If the specimen has not been damaged after specified fatigue times, check whether the residual strength of the structure is still within the design range. Therefore, the secondary fatigue damage is to consider the mechanical behavior of composite stiffened panels in the full life cycle.

Point4: Why not conduct the micro-performance analysis and tests after the impact and fatigue tests, such as scanning electron microscope? This is very effective for revealing the damage and failure mechanism of materials, such as matrix cracking, fiber fracture, interface debonding and delamination damage, etc.

Response 4: If the test piece can be analyzed by scanning electron microscope, the damage and failure mechanism analysis of the material should be more clear. However, the stiffened panel is a structure component, which needs to be cut locally to make scanning electron microscope analysis, which often destroys the original morphology and cannot obtain the desired results. Generally, the failure mode can be seen from the fracture surface of the test piece.

Point5: The third part should focus on the finite element simulation considering the failure criterion and damage evolution. Therefore, the current subtitle may not be appropriate and needs further adjustment or modification.

Response 5: The title of the third part has been changed to CAI progressive damage analysis.

Point6: In figure 7, it is difficult to see the impact damage mode clearly. Do you have some local magnified pictures to clearly show the damage mode? In addition to Table 6, the results and related analysis on the impact strength and fracture energy are suggested.

Response 6: Impact damage is mainly internal damage. Generally, the area of damage is detected by NDT technology, and the type of damage cannot be determined. It can be seen from Figure 7 that there is almost no damage on the surface, and the damage area is within the white circle. Because the specimens are all subjected to 50J energy, and the number is limited, it is impossible to analyze the impact strength and fracture energy.

Point7: In table 7 and figure 10, the maximum buckling and failure load are more than 1000kN. On the other hand, the compression tests were conducted on an hydraulic fatigue testing machine with a 1000 kN load sensor. Whether there are some contradictions in these two parts, it is suggested to provide relevant explanations.

Response 7: Here is our mistake. We used a 500T MTS tester. The test cannot be completed on a 10000KN testing machine.

Point8: In figure 11(a), for the load-displacement curves of C1, the initial load is 0 with the displacement. So it is recommended to further adjust the curves by moving to the left to analyze the relationship between some simulation and experimental results.

Response 8: The load displacement curve in Fig.11a has been adjusted. The corresponding simulation and test comparison are supplemented in Section 4.3.

Point9: Conclusions should be further improved including 3 to 4 key points.

Response 9: The conclusions are further sorted out and 4 items are summarized.

Reviewer 2 Report

we added the revision in the attachment.

Author Response

Response to Reviewer 2 Comments

Point 1: The abstract should include the summarized results and the trend with highlight the important postulate that every study agrees with. Moreover, the most important value from tests also mentioned. 

Response 1: The summary has been reorganized and important conclusions in the article have been listed. 

Point 2: Before it goes to the VID or BVID, the general postulates that mentioned in the first paragraph should be strengthen with the references. Please used the following references to increase the quality of the introduction: 10.1016/j.compositesb.2019.05.049, and 10.3390/polym14204322

Response 2: The recommended references have been cited in the text.

Point 3: Please add the abbreviation of the USDFLD?

Response 3: The full name has been given in the paper.

Point 4: If the study have purposes to compare the 4 model, why the shape and the dimension of each model have differences? The different shape and dimension can trigger the different results. How do you elaborate this sample preparation?

Response 4: The purpose of this paper is to study the post buckling ratio, buckling and post buckling analysis methods, and the influence of fatigue. It is not necessary to draw conclusions directly from the test, so the differences in size and shape in the test do not affect the conclusions. If there are a large number of test pieces, each variable can be tested.

Point 5: The impact study was the best when it takes the used of high impact load. However, in the present study, the authors state that they used low-velocity impact load. Why they used these methods? And then why not compared with high velocity impact load?

Response 5: Low speed impact itself is a dynamic problem, which can be studied by dynamic analysis method. For high-speed problems, it is more difficult to solve because of geometric nonlinearity and material nonlinearity. For the problem that can be calculated with high speed impact, it can certainly solve the low speed problem. The comparison between low speed impact and high speed impact can be carried out in subsequent research.

Point 6: The previous question also mentioned. Why used the slow load rate? Why not used several load rate such as 0.5, 2, 4, etc and then compared it!

Response 6: The specimens in this paper are limited, and the effect of loading rate is not considered. Moreover, according to the author's research on material test pieces, it is found that the loading rates of 0.5, 2 and 4 have little effect on the test results, and they all belong to static loading.

Point 7: The failure criterion by using Hashin is based on the max load or strength of the materials. How the calculation from experiment can be applied with the simulation analysis? Please add the following references to strengthen the findings. 10.1016/j.compstruct.2021.113707.

Response 7: It has been pointed out in Section 3.1 that the strength parameters are obtained through unidirectional plates tests and the corresponding references are cited.

Point 8: The mesh ratio and mesh quality should be addressed in the manuscript to make the readers easier to understand.

Response 8: The finite element in this paper adopts S4R element, the mesh size is 5mm, and the total number of meshes is 70530, which has been described in this paper.

Point 9: The impact test was combined with NDT. However, the NDT was state with “NDT were conducted after impact test, and damage area are recorded in Table 6”. How the authors can determine the study related to the NDT results?

Response 9: A scan was carried out on the specimens after impact to determine the damage area, and the corresponding statement has been added in Section 2.2 of the article.

Point 10: More questions related to the damage area. How was the areas determined as like in table 6? How the author can draw the damage area as shown in white line?

Response 10: The definition of damage area has indicated in the damage photos and described in Section 4.1.

Point 11: Please revise the conclusion. Add all the best results in value mode of all tests. Collect all the narration into a single paragraph. If necessary, make point to point based of all the results of each tests and simulation process.

Response 11: The conclusions were reorganized, and each research content summarized one conclusion, a total of four conclusions.

Round 2

Reviewer 1 Report

The revised version has responded well all comments, it can be accepted.

Reviewer 2 Report

The present paper is suitable to be published.